# A Shared Frailty Model for Left-Truncated and Right-Censored Under-Five Child Mortality Data in South Africa

Tshilidzi Benedicta Mulaudzi [1], Yehenew Getachew Kifle [2],*, and Roel Braekers [3]

1 Department of Mathematical and Computational Sciences, University of Venda, Thohoyandou 0950, South Africa; tshilidzi.mulaudzi@univen.ac.za
2 Department of Mathematics and Statistics, University of Maryland, Baltimore County, MD 21250, USA
3 Data Science Institute, Center for Statistics, Hasselt University, 3500 Diepenbeek, Belgium; roel.braekers@uhasselt.be
* Correspondence: yehenew@umbc.edu; Tel.: +1-410-455-2972

**Abstract:** Many African nations continue to grapple with persistently high under-five child mortality rates, particularly those situated in the Sub-Saharan region, including South Africa. A multitude of socio-economic factors are identified as key contributors to the elevated under-five child mortality in numerous African nations. This research endeavors to investigate various factors believed to be associated with child mortality by employing advanced statistical models. This study utilizes child-level survival data from South Africa, characterized by left truncation and right censoring, to fit a Cox proportional hazards model under the assumption of working independence. Additionally, a shared frailty model is applied, clustering children based on their mothers. Comparative analysis is performed between the results obtained from the shared frailty model and the Cox proportional hazards model under the assumption of working independence. Within the scope of this analysis, several factors stand out as significant contributors to under-five child mortality in the study area, including gender, birth province, birth year, birth order, and twin status. Notably, the shared frailty model demonstrates superior performance in modeling the dataset, as evidenced by a lower likelihood cross-validation score compared to the Cox proportional hazards model assuming independence. This improvement can be attributed to the shared frailty model's ability to account for heterogeneity among mothers and the inherent association between siblings born to the same mother, ultimately enhancing the quality of the study's conclusions.

**Keywords:** survival; under-five child mortality; Cox PH hazards model; frailty model; right censored; left truncation

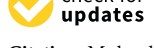



## 1. Introduction

Under-five child mortality (U5CM) remains a significant challenge in Sub-Saharan Africa. The probability of children under five years old dying in Sub-Saharan Africa is fourteen times higher than that of children in developed regions worldwide [1]. Millennium Development Goal 4 (MDG-4) aimed to reduce the under-five child mortality by two-thirds between 1990 and 2015 [2]. Furthermore, the Sustainable Development Goals (SDGs) seek to eliminate under-five child mortality and deaths of newborn babies by 2030. Globally, efforts are focused on reducing the mortality rate of newborns within their first 28 days of life to less than 12 per 1000 live births, as well as decreasing the mortality rate of children under five to less than 25 per 1000 live births [3].

South Africa, situated within Sub-Saharan Africa, is actively working towards reducing under-five mortality rates in alignment with the SDG targets. To achieve this goal, policymakers and healthcare authorities require comprehensive insights into the causes of child deaths, enabling them to monitor child health and service provision effectively [4]. Consequently, under-five mortality has garnered substantial attention from researchers worldwide, prompting investigations into its diverse determinants. Many studies have employed a range of

statistical regression models, such as logistic regression and Cox proportional hazards models, primarily to explore the relationships between various risk factors and child mortality.

In the context of time-to-death analysis, logistic regression falls short as an optimal choice due to its exclusive reliance on the binary event indicator, disregarding detailed time-to-event information [5]. Conversely, the Cox PH model leverages all available information, encompassing time-to-event and censoring indicators. However, when applied under the assumption of independence among individuals' survival times, this model may not hold in cases involving clustered survival data. Typically, observations within the same cluster share unobservable characteristics, resulting in correlated outcomes. Despite a consensus among analysts that such associations should be considered in survival analysis, they are often overlooked, potentially yielding inefficient and biased estimates [6]. Furthermore, neglecting these associations within clusters can lead to inaccuracies in standard error estimates. For instance, children within the same family or with the same mother often share environmental and genetic factors and experience similar parental care and socioeconomic conditions. Covariates shared among children of the same mother introduce correlations in their mortality risks. Ignoring these connections when estimating risks among siblings can yield erroneous results [7].

This study is motivated by a dataset on the clustered (at the mother level) right-censored, left-truncated time to death of children under five in South Africa. Many of such U5CM datasets are analyzed using the classical univariate Cox PH model which assumes that the times to death of children from the same mother are independent of each other, which is not the case in reality. Several models have been introduced in the literature to take such an association in clustered data into account [8,9]. As the first model, we consider a marginal Cox PH model under an independent working condition. This model was introduced in [10] and assumes that the parameter estimates are found under the assumption that the event times in a cluster are independent. Afterwards, the standard errors are corrected for this association by a robust estimator. Frailty models, on the other hand, are extensions of the Cox proportional hazard model, which is the most popular regression technique for time-to-event data [11]. The frailty approach is a statistical modeling concept that accounts for the heterogeneity in the model. Thus, a frailty model is a random effect model for time-to-event data, where the random effect, also known as "frailty", has a multiplicative effect on the baseline hazard function [12,13].

A shared frailty model is a random effect model where the frailties are common (or shared) among study subjects within a cluster [14,15]. Therefore, this frailty model will account for the heterogeneity which is related to the event of interest. In this paper, we have applied this shared frailty model to the U5CM left-truncated dataset, with the assumption that children from the same mother (cluster) share similar risk factors, and further compared this model with both the classical univariate Cox PH model and the marginal Cox PH model under working independence conditions. The likelihood cross-validation (*LCV*) criterion is used to identify and choose the better-performing model among the univariate and clustered Cox PH models [16].

## 2. Materials and Methods

### 2.1. Data Source and Study Variables

This study utilized two distinct datasets sourced from Statistics South Africa (Stats SA): the "mortality and causes of death" dataset and the "recorded live birth" dataset. To merge these datasets, several matching variables were employed, including birth province, birth year, birth month, gender, and more. The resultant merged dataset contains pertinent information concerning children born between 2010 and 2015. However, it is worth noting that this dataset has left truncation due to missing data in the original Stats SA datasets, for which the reasons were not provided by Stats SA.

To streamline the dataset, children born outside the nine provinces of South Africa and those lacking maternal identity information were excluded. The dataset was structured into clusters based on maternal identity numbers, with each mother representing a unique

cluster. Additionally, clusters comprising only a single member were omitted from the dataset to align with our primary focus on analyzing the survival durations of siblings while accounting for unobservable familial risk factors (i.e., those associated with the same mother). In total, the refined dataset encompasses 250,260 children distributed across 123,110 distinct clusters (mothers). Using this final dataset, we computed the response variable, which denotes *the time to either death or censoring in days*. This calculation spans from the child's date of birth to the date of either death or censoring if the child survived beyond the study period.

The outcome variable for this study is the survival time of children, measured in days. Children who remain alive at the conclusion of the study period are considered right-censored. Among the various covariates considered to influence the survival of children under the age of five, the study includes factors such as gender, birth year, birth province, birth order, and twin status. A comprehensive list of the variables employed in this study is presented in Table 1.

**Table 1.** List of variables used in the study.

| Variables | | |
|---|---|---|
| **Name** | **Description** | **Category** |
| Gender | Child gender | Female or Male |
| Province | Birth province | Limpopo, Eastern Cape, Free State, Gauteng KwaZulu Natal, Mpumalanga, North West Northern Cape, Western Cape |
| Year | Birth year | 2009–2015 |
| Twin | Twin identifier | Belonging to a twin pair or not |
| Birth order | Previous number of living children | Eldest, second, third, fourth, fifth, etc. |
| Status | Survival status indicator | Dead or alive |
| Time | Follow-up time (in days) | Duration in days between birth date and day of death or censoring |
| TrunTime | Time for left truncation | Number of days between birth date and day of truncation (31 December 2012) |
| ClusterId | For distinguishing siblings born to the same mother | |

Dataset

Figure 1 depicts the U5CM data collection scheme over time. Due to the nature of the data, we cannot have death information for those born between 2010 and 2012. If a child died between 2010 and 2012, they are missed or not observable, see "Not observed" in Figure 1. For a child born between 2010 and 2012 to be observable, he/she should survive beyond 2012. Those observed samples born between 2010 and 2012 are defined as "truncated samples" that are subject to $T \geq \tau$, where $T$ is death time and $\tau$ is truncation time. Those children born after 2012 are always observable. Besides left-truncation, samples may be right-censored by the end of the data collection, 2015. For those censored cases, we only know that the death time is greater than the observed censoring time $C$ (the duration of survival up to 2015). Please refer to Table 2 for a descriptive summary of the U5CM dataset.

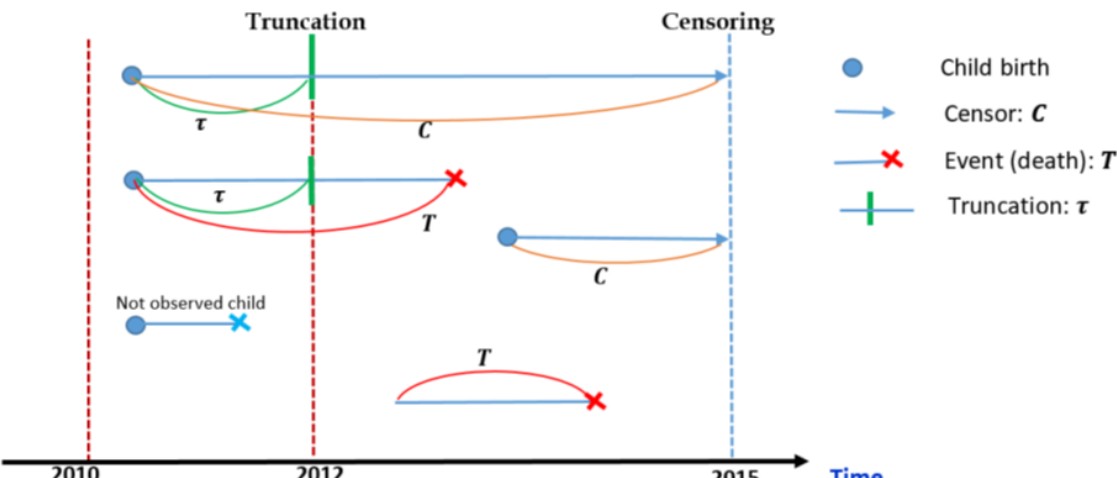

**Figure 1.** Left-truncated and right-censored U5CM data. **T** is event time (time from birth to death), $\tau$ is truncation time (time from birth to 2012), and **C** is censoring time (time from birth to 2015).

**Table 2.** Descriptive summary of the South African under-five child mortality dataset.

| Factors | Levels | Total (%) | Death N (%) | Censored N (%) | Left Truncated N (%) |
|---|---|---|---|---|---|
| Gender | Female | 124,901 (49.9%) | 2482 (2.0%) | 122,419 (98.0%) | 15,282 (50%) |
| | Male | 125,359 (50.1%) | 2720 (2.2%) | 122,639 (97.8%) | 15,274 (50%) |
| Province | Limpopo | 29,779 (11.9%) | 824 (2.8%) | 28,955 (97.2%) | 3189 (10.4%) |
| | Eastern Cape | 33,146 (13.2%) | 598 (1.8%) | 32,548 (98.2%) | 4761 (15.6%) |
| | Free State | 11,299 (4.5%) | 402 (3.6%) | 10,897 (96.4%) | 997 (3.3%) |
| | Gauteng | 51,328 (20.5%) | 1047 (2.0%) | 50,281 (98.0%) | 4812 (15.7%) |
| | KwaZulu | 59,876 (23.9%) | 766 (1.3%) | 59,110 (98.7%) | 9425 (30.8%) |
| | Mpumalanga | 21,073 (8.4%) | 458 (2.2%) | 20,615 (97.8%) | 2705 (8.9%) |
| | North West | 15,530 (6.2%) | 517 (3.3%) | 15,013 (96.7%) | 2431 (8.0%) |
| | Northern Cape | 5782 (2.3%) | 249 (4.3%) | 5533 (95.7%) | 1703 (5.6%) |
| | Western Cape | 22,447 (9.0%) | 341 (1.5%) | 22,106 (98.5%) | 533 (1.7%) |
| Year | 2010 | 6887 (2.8%) | 11 (0.2%) | 6876 (99.8%) | 6887 (22.5%) |
| | 2011 | 8676 (3.5%) | 24 (0.3%) | 8652 (99.7%) | 8676 (28.4%) |
| | 2012 | 14,993 (6.0%) | 189 (1.3%) | 14,804 (98.7%) | 14,993 (49.1%) |
| | 2013 | 103,811 (41.5%) | 3624 (3.5%) | 100,187 (96.5%) | 0 (0%) |
| | 2014 | 12,694 (5.1%) | 191 (1.5%) | 12,503 (98.5%) | 0 (0%) |
| | 2015 | 103,199 (41.2%) | 1163 (1.1%) | 102,036 (98.9%) | 0 (0%) |
| Twin | Yes | 197,956 (79.1%) | 3775 (1.9%) | 194,181 (98.1%) | 28,781 (94.2%) |
| | No | 52,304 (20.9%) | 1427 (2.7%) | 50,877 (97.3%) | 1775 (5.8%) |
| Order | 0 (No ordering) | 147,274 (58.8%) | 4073 (2.8%) | 143,201 (97.2%) | 27,525 (90.1%) |
| | 1 | 100,424 (40.1%) | 1098 (1.1%) | 99,326 (98.9%) | 3000 (9.8%) |
| | 2 | 2504 (1.0%) | 29 (1.2%) | 2475 (98.8%) | 31 (0.1%) |
| | 3 | 55 (0.0%) | 2 (3.6%) | 53 (96.4%) | 0 (0%) |
| | 4 | 3 (0.0%) | 0 (0.0%) | 3 (100.0%) | 0 (0%) |

*2.2. Methods of Data Analysis*

2.2.1. Cox Proportional Hazards Model

The predominant regression model employed in many biomedical studies for analyzing right-censored survival data is the Cox proportional hazards (PH) model, which was introduced by David Cox in 1972 [11]. Within the context of the *frailtypack* software package, it is possible to employ the Cox PH model, with parameter estimation carried out through a penalized likelihood estimation approach [17]. This classical Cox PH model is

marginal in nature and does not account for clustering structures present in the dataset. Instead, it treats all survival times within the same cluster as independent events [8].

To illustrate the Cox PH model, let us consider the hazard function at time $t$ for the $j^{th}$ individual (where $j = 1, 2, \ldots, n$) with a covariate vector $\mathbf{X}_j$, which can be formulated as follows:

$$h_j(t) = h_0(t) \, exp(\boldsymbol{\beta}' \mathbf{X}_j), \tag{1}$$

where $h_0(t)$ denotes the baseline hazard function at time $t$ and $\boldsymbol{\beta}$ represents the regression coefficients associated with the covariate vector $\mathbf{X}_j$. The baseline hazard function $h_0(t)$ describes the hazard function for an individual when all covariate values are set to zero.

The model parameters are estimated through the Cox partial likelihood approach, wherein only the probabilities of individuals who experience an event are taken into account. The partial likelihood can be expressed as follows:

$$L_{partial}(\boldsymbol{\beta}) = \prod_{i=1}^{r} \frac{exp(\boldsymbol{\beta}' \mathbf{X}_{(i)})}{\sum_{l \in R(t_{(i)})} exp(\boldsymbol{\beta}' \mathbf{X}_l)} \tag{2}$$

where $\mathbf{X}_{(i)}$ represents the vector of covariates for the subject with the $i^{th}$ ordered event time, and $R(t_{(i)})$ denotes the risk set at the $i^{th}$ ordered event time. Consequently, the sum in the denominator encompasses all subjects who are still at risk at time $t_{(i)}$. Note that censored observations contribute to this denominator only, reflecting their impact solely through this part of the equation [18].

The distinction between the classical Cox PH model and the marginal Cox PH model lies in the way that the standard errors of the parameter estimates are estimated. In the classical Cox PH model, we assume that the event times are independent and estimate the standard errors in that way. To take the association into account, the marginal Cox model considers a robust version of the standard errors. This model assumes that different survival times are linked to one another in a cluster and does not specify the size of the association. It only corrects for this.

### 2.2.2. Shared Frailty Model

The shared frailty model is essentially a variation of the Cox proportional hazards (PH) model that incorporates a random effect. This random effect is introduced within each cluster, implying that individuals within the same cluster exhibit more similarity than those belonging to different clusters. Furthermore, this random effect provides insight into the unobservable factors that affect all individuals within a given cluster [14,19]. In our study, we specifically deal with left-truncated and right-censored data, which are the relevant data types for our research context.

In our specific context, let us suppose we have a total of $n$ children coming from $k$ distinct clusters (mothers). We collect data denoted by $Y_{ij}$, which corresponds to the minimum of $T_{ij}$ and $C_{ij}$, along with indicators that signify censoring status, denoted as $\delta_{ij}$. Specifically, $\delta_{ij}$ takes the value 1 if $T_{ij} \leq C_{ij}$ and 0 otherwise. In this context, $T_{ij}$ represents the survival times, while $C_{ij}$ represents the censoring times for the individuals under investigation. We classify survival times as left-truncated when our observations encompass only individuals for whom $T_{ij} > L_{ij}$. It is important to note that we assume that $T_{ij}$, $C_{ij}$, and $L_{ij}$ are independent of one another.

The shared frailty model is given by:

$$h_{ij}(t|z_i) = z_i h_0(t) exp(\boldsymbol{\beta}' \mathbf{X}_{ij}), \tag{3}$$

where $h_0(t)$ represents the baseline hazard at time $t$, $\mathbf{X}_{ij}$ stands for the covariate vector for individual $j$ within cluster $i$, $\boldsymbol{\beta}$ represents a vector of regression coefficients, and $z_i$ represents frailties, which are independently and identically distributed from a gamma distribution with a mean of 1 and an unknown variance denoted as $\theta$.

In this context, it is important to note that higher frailty values for children imply greater frailty, and consequently, an expectation of a shorter time to the event of interest compared to individuals with equivalent measured covariates [20]. When employing shared frailty models, larger frailty values ($z_i > 1$) indicate that the event is more likely to occur earlier, in contrast to clusters with smaller frailty values ($z_i < 1$) [21]. It is crucial to emphasize that there is no correlation between frailties across different clusters, but there is an association among individuals within the same cluster [22]. In the context of positive outcomes, such as pregnancy or recovery, subjects with higher "frailty" are expected to achieve the positive outcome sooner than others with the same set of covariates.

## 3. Results

As shown in Table 2, a total of 250,260 children and 123,110 mothers (clusters) were included in the analysis. Out of the total cohort of 250,260 children, 5202 (2.1%) had experienced mortality, while 245,058 (97.9%) remained alive at the end of the follow-up period. The child mortality rates within South Africa's provinces exhibited variability, with the highest observed in the Northern Cape (4.3%) and the Free State (3.6%), while the lowest occurred in KwaZulu-Natal (1.3%) and the Western Cape (1.5%). Regarding gender, a higher percentage of male children (2.2%) experienced mortality compared to their female counterparts (2.0%). Across birth years, the highest mortality rate (3.5%) was recorded in 2013, contrasting with the lowest rate (0.2%) in 2010. Children born as twins had a lower mortality rate than singletons, with 1.9% of twins and 2.7% of singletons experiencing mortality before the age of five. Furthermore, the majority of children in the dataset were censored.

The mortality rate also displayed variations based on the number of prior children born to mothers (birth order). Specifically, mothers with four previous living children had the lowest mortality rate (0.0%). This suggests that greater maternal experience, reflected in a higher number of prior births, was associated with a reduced risk of child mortality, as noted by Srivastava et al. (2021) [23].

Regarding left-truncated individuals, a total of 30,556 (12.2%) were found to be left-truncated. Half of them were females, and the other half were males. The highest proportion of left-truncated individuals was observed in KwaZulu-Natal province (30.8%), followed by Gauteng (15.7%) and Eastern Cape (15.6%), with the lowest in Western Cape (1.7%). Across birth years, the highest percentage of left-truncated individuals (49.1%) occurred in 2012, with no instances recorded between 2013 and 2015. Additionally, the dataset included a higher proportion of left-truncated twins (94.2%) compared to singletons (5.8%). Concerning the number of previous living children that mothers had (birth order), the highest proportion of left-truncated individuals (90.1%) was observed when mothers had no prior living children.

We employed the *frailtypack* package in R to apply the univariate Cox, marginal Cox, and gamma shared frailty models. Initially, we fitted a Cox model under the working assumption of independence within clusters. Subsequently, to account for the associations, we also fitted a Cox model using robust standard errors with our dataset. The outcomes of both Cox models are detailed in Table 3.

Table 3 displays the prospective risk factors associated with the high under-five child mortality rate in South Africa. The model parameters were determined through a penalization approach proposed by Rondeau et al. (2012) [17]. It is noteworthy that while the robust standard errors are marginally larger than the regular standard errors, the coefficient estimates and *p*-values remain unchanged.

**Table 3.** Results from the Cox PH models with both unadjusted and robust standard errors.

| Factors | Levels | Coef | Hazard Ratio HR | Unadjusted SE (Robust SE) | *p*-Value |
|---|---|---|---|---|---|
| Gender | *Female (Ref)* | | | | |
| | Male | 0.0946 | 1.0992 | 0.0278 (0.0280) | 0.0007 |
| Province | *Limpopo (Ref)* | | | | |
| | Eastern Cape | −0.3899 | 0.6771 | 0.0538 (0.0552) | <0.0001 |
| | Free State | 0.2512 | 1.2856 | 0.0609 (0.0622) | <0.0001 |
| | Gauteng | −0.3311 | 0.7181 | 0.0466 (0.0477) | <0.0001 |
| | KwaZulu | −0.7372 | 0.4785 | 0.0520 (0.0533) | <0.0001 |
| | Mpumalanga | −0.2208 | 0.8019 | 0.0583 (0.0597) | 0.00022 |
| | North West | 0.2594 | 1.2962 | 0.0562 (0.0569) | <0.0001 |
| | Northern Cape | 0.4782 | 1.6131 | 0.0724 (0.0746) | <0.0001 |
| | Western Cape | −0.6318 | 0.5317 | 0.0644 (0.0650) | <0.0001 |
| Year | *2010 (Ref)* | | | | |
| | 2011 | 0.8213 | 2.2731 | 0.293 (0.3102) | 0.0051 |
| | 2012 | 1.2830 | 3.6090 | 0.324 (0.3340) | <0.0001 |
| | 2013 | 1.8350 | 6.2644 | 0.342 (0.3520) | <0.0001 |
| | 2014 | 1.4220 | 4.1460 | 0.325 (0.3622) | <0.0001 |
| | 2015 | 1.6871 | 5.4060 | 0.346 (0.3500) | <0.0001 |
| Twin | *No (Ref)* | | | | |
| | Yes | 0.1672 | 1.1820 | 0.0345 (0.0363) | <0.0001 |
| Order | *No (Ref)* | | | | |
| | ≥1 | −0.3667 | 0.6929 | 0.0475 (0.0506) | <0.0001 |
| | Likelihood ratio test | 1075 (*p* < 0.0001) | | | |
| | Wald test | 1047 (*p* < 0.0001) | | | |
| | Score test | 1096 (*p* < 0.0001) | | | |
| | Penalized marginal log-likelihood | −55,833.07 | | | |
| | LCV | 0.2232 | | | |

In this analysis, we have incorporated potential factors expected to influence children's survival, including the child's gender, birth province, birth year, twin status, and birth order. Notably, male children were found to have a significantly higher risk of mortality compared to females, with a hazard ratio (HR) of 1.1. The hazard ratios for several provinces, namely Eastern Cape, Gauteng, KwaZulu Natal, Mpumalanga, and Western Cape, were all below 1, signifying that children residing in these provinces were less prone to mortality than those in the reference province, Limpopo. Furthermore, the results unveiled that children born between 2011 and 2015 faced a higher risk of mortality compared to those born in 2010. Table 3 also presents the hazard of death for children who were one of a twin versus singletons. The Cox model outcomes indicate that children born in a set of twins had a significantly higher hazard of death (HR = 1.1), suggesting that twins were more vulnerable to mortality than singletons. Birth order emerged as another significant factor influencing under-five child mortality in South Africa. Children born to mothers with previous children in the family exhibited a reduced hazard of death, with a hazard ratio of 0.690. This finding implies that a higher number of children born to the mother in the past was associated with a lower risk of death for the current child. To assess the overall significance of the covariates, we employed three test statistics, the likelihood ratio test, Wald test, and the score test, as presented in Table 3. Notably, all three statistics yielded very small *p*-values (*p* < 0.0001), signifying their high statistical significance. This indicates that at least one of the covariates significantly contributes to the mortality of children under five in South Africa.

To address the potential cluster effect at the mother's level, we also applied a marginal Cox PH model. In this model, the robust standard errors for various covariates were slightly larger than those in the univariate Cox PH model. This suggests that while the

individual covariates remain significant, the association between event times within a cluster has a limited impact. Additionally, we employed a shared frailty model to quantify the heterogeneity among different clusters, thus enhancing our ability to identify potential risk factors associated with child mortality in the study area (refer to Table 4). We utilized the same set of covariates as in the previous section where we fitted the Cox PH models. The findings from the shared frailty model closely paralleled those obtained from the Cox model. All covariates (except for the birth year 2011 compared to 2010) were identified as significant factors for under-five child mortality in South Africa. Notably, female children exhibited a lower likelihood of mortality compared to male children, and children residing in provinces other than Limpopo were less likely to experience mortality. Furthermore, the shared frailty model revealed that children born between 2011 and 2015 faced a higher likelihood of mortality than those born in 2010. Twins were also found to be at a higher risk of mortality compared to singletons. Additionally, a lower hazard of mortality was observed for children who were not first-born children. To assess the significance of the frailty (clustering) effect in our model, we evaluated the value of $\theta$, as shown in Table 4. The variance of the frailty term ($\theta = 2.342$) with a *p*-value of $<0.0001$ indicates the presence of significant clustering (heterogeneity) at the mother's level. This significant frailty term captures the influence of factors or covariates not included in the model. Furthermore, the shared frailty model yielded a lower likelihood cross-validation value ($LCV = 0.2228$) compared to the Cox PH models ($LCV = 0.2232$), suggesting that the shared frailty model, with mothers as clusters, is a superior model compared to the marginal Cox PH model, where all children are treated as independent of each other.

**Table 4.** Results from the shared frailty model (clustered Cox proportional hazard model).

| Factors | Levels | Shared Frailty Model | | | |
|---|---|---|---|---|---|
| | | Coef | Hazard Ratio | Standard Error | *p*-Value |
| Gender | *Female (Ref)* | | | | |
| | Male | 0.096 | 1.100 | 0.029 | 0.0010 |
| Province | *Limpopo (Ref)* | | | | |
| | Eastern Cape | −0.404 | 0.667 | 0.062 | <0.0001 |
| | Free State | 0.268 | 1.308 | 0.070 | 0.0001 |
| | Gauteng | −0.336 | 0.714 | 0.054 | <0.0001 |
| | KwaZulu | −0.758 | 0.468 | 0.055 | <0.0001 |
| | Mpumalanga | −0.226 | 0.798 | 0.066 | 0.0007 |
| | North West | 0.270 | 1.310 | 0.066 | <0.0001 |
| | Northern Cape | 0.517 | 1.678 | 0.083 | <0.0001 |
| | Western Cape | −0.646 | 0.524 | 0.071 | <0.0001 |
| Year | *2010 (Ref)* | | | | |
| | 2011 | 0.824 | 2.279 | 0.497 | 0.0972 |
| | 2012 | 1.288 | 3.626 | 0.192 | <0.0001 |
| | 2013 | 1.883 | 6.577 | 0.175 | <0.0001 |
| | 2014 | 1.446 | 4.247 | 0.197 | <0.0001 |
| | 2015 | 1.723 | 5.603 | 0.180 | <0.0001 |
| Twin | *No (Ref)* | | | | |
| | Yes | 0.180 | 1.197 | 0.037 | <0.0001 |
| Order | *0 (Ref)* | | | | |
| | ≥1 | −0.372 | 0.690 | 0.049 | <0.0001 |
| $\theta$ (*p*-value) | | 2.342 ($p < 0.0001$) | | | |
| Penalised marginal log-likelihood | | −55,742.73 | | | |
| LCV | | 0.2228 | | | |

## 4. Discussion

Our primary objective in this study was to assess the influence of potential factors on under-five child mortality in South Africa, with a particular focus on addressing the clustering effect and left truncation in the dataset. We applied various survival time-to-death models, both with and without considering clustering at the mother's level.

Our findings indicate that the mortality of children under the age of five is influenced by their gender, birth province, birth year, birth order, and whether they are part of a set of twins. Notably, boys were found to have a higher likelihood of mortality than girls, consistent with findings from studies conducted in Bangladesh [24], Uganda [25], Ethiopia [26], Nigeria [27], and Turkey [28]. Research in [24,29], among others, has suggested that female children possess a biological advantage against many causes of mortality compared to male children. Furthermore, our results revealed that twins are more susceptible to mortality than singletons, a finding consistent with the work of researchers like in [26,30]. Additionally, first-born children were at a higher risk of mortality compared to those born as second, third, fourth, or fifth children. This trend might be attributed to the experience mothers gain in caring for children over time. The study also highlighted that children born in provinces such as Eastern Cape, Gauteng, KwaZulu Natal, and Western Cape had a lower risk of mortality compared to those born in Limpopo. This disparity may be due to Limpopo's predominantly rural setting, where access to healthcare facilities is limited.

The calculated value of $\theta$, the variance of the frailty term, suggests the existence of unobserved heterogeneity at the mother's level, indicating the presence of additional factors contributing to child mortality beyond those considered in the model. The shared frailty model, which accounts for this unobserved variation, exhibited slightly increased regression parameter estimates compared to the univariate Cox proportional hazards model. This is primarily because the shared frailty model accommodates the extra variance associated with unmeasured or unaccounted-for risks. Moreover, we observed that the effects of covariates included in the model were biased downward when frailty effects were not considered, aligning with the findings of Liu et al. (2004) [31]. Lastly, the lower likelihood cross-validation (*LCV*) value of the shared frailty model, compared to the Cox PH model, indicates that the shared frailty model provides a better fit for the under-five child mortality dataset.

## 5. Conclusions

In this study, we explored the factors associated with under-five child mortality using both Cox PH and shared frailty models. We employed a penalized likelihood estimation technique for hazard function estimation in both models. The findings underscored the significance of gender, birth province, birth year, twin status, and birth order as crucial determinants of under-five child survival in South Africa. To enhance child survival in South Africa, targeted interventions are warranted, particularly for first-time mothers. These interventions should focus on improving access to healthcare services, educating first-time mothers on child care practices, and providing special attention to twins during their first year of life. Such measures can contribute to a reduction in under-five child mortality in the study area.

Moreover, our results indicated that the shared frailty model outperformed the Cox PH model, as evidenced by the lower likelihood cross-validation (*LCV*) results. This preference for the shared frailty model was driven by the positive correlation observed in the dataset, highlighting the necessity of accounting for data clustering in clustered time-to-event models. The presence of heterogeneity among mothers and a strong association among children from the same mother, as identified in the shared frailty model, underscore the importance of incorporating clustering considerations in modeling. This approach ultimately enhanced the robustness and accuracy of our study's findings.

*Study Limitation*

Our dataset contains a substantial number of children with clusters consisting of just one individual; thus, this poses challenges when applying the shared frailty model. While

these isolated individuals may not significantly contribute to our primary objective, which is analyzing clustered left-truncated, right-censored data while considering unobserved risk factors within children sharing the same mother, we suggest incorporating these cases with a cluster size of one using an alternative modeling approach. Such an approach should be capable of handling high-dimensional and unbalanced clusters, such as copula survival models.

**Author Contributions:** T.B.M., with guidance from Y.G.K. and R.B., undertook various tasks, such as secondary data collection, data cleaning, and statistical analysis. T.B.M. also took the lead in drafting the manuscript. Y.G.K. and R.B. were responsible for conceptualizing and designing the study, as well as overseeing project coordination. They provided critical input during the manuscript review process. All authors have read and agreed to the published version of the manuscript.

**Funding:** This work in the context of a PhD project was supported by grants from the Institutional University Cooperation IUC-UL (University of Limpopo) project under the umbrella of the Flemish Interuniversity Council (VLIR-UOS).

**Institutional Review Board Statement:** Since the data utilized in this study consist of publicly available secondary data from the Statistics South Africa (Stats SA) website, no ethical clearance or approval was necessary for the investigation.

**Informed Consent Statement:** This study is based on readily accessible secondary data sourced from Statistics South Africa. Written informed consent has been obtained during the primary data collection phase by Stats SA.

**Data Availability Statement:** This study is based on readily accessible secondary data sourced from Statistics South Africa (www.statssa.gov.za).

**Acknowledgments:** The authors would like to acknowledge Stats SA for providing the U5CM dataset. Moreover, we extend our sincere gratitude to the anonymous reviewers for their great comments, which significantly improved our manuscript.

**Conflicts of Interest:** The authors declare no conflict of interest.

## Abbreviations

The following abbreviations are used in this manuscript:

| | |
|---|---|
| AIC | Akaike Information Criterion |
| HR | Hazard ratio |
| LCV | Likelihood cross-validation |
| MDG | Millennium Development Goal |
| PH | Proportional hazards |
| SDG | Sustainable Development Goal |
| Stats SA | Statistics South Africa |
| U5CM | Under-five child mortality |

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
