# Peer review of "A Shared Frailty Model for Left-Truncated and Right-Censored Under-Five Child Mortality Data in South Africa"

_stats, doi:10.3390/stats6040063_

Round 1

Reviewer 1 Report

Read recent articles relevant to your topic; check how many references other authors could be included in their papers for the same article type as yours.

To give you a general idea, the following are some examples:

 Rows 38-40 - Reference:

  1. Zhang H, Schaubel DE, Kalbfleisch JD. Proportional hazards regression for the analysis of clustered survival data from case-cohort studies. Biometrics. 2011 Mar;67(1):18-28. doi: 10.1111/j.1541-0420.2010.01445.x. PMID: 20560939; PMCID: PMC4458467.
  2. de Jong VMT, Moons KGM, Riley RD, Tudur Smith C, Marson AG, Eijkemans MJC, Debray TPA. Individual participant data meta-analysis of intervention studies with time-to-event outcomes: A review of the methodology and an applied example. Res Synth Methods. 2020 Mar;11(2):148-168. doi: 10.1002/jrsm.1384. Epub 2020 Feb 6. PMID: 31759339; PMCID: PMC7079159.

 Rows 82-83 - Reference:

Peng Y, Taylor JM, Yu B. A marginal regression model for multivariate failure time data with a surviving fraction. Lifetime Data Anal. 2007 Sep;13(3):351-69. doi: 10.1007/s10985-007-9042-4. Epub 2007 Jul 20. PMID: 17641970.

Row 59 - Reference:

Shu Kay Ng, Richard Tawiah, Geoffrey J Mclachlan, Vinod Gopalan, Joint frailty modeling of time-to-event data to elicit the evolution pathway of events: a generalized linear mixed model approach, Biostatistics, Volume 24, Issue 1, January 2023, Pages 108–123, https://doi.org/10.1093/biostatistics/kxab037

Rows 69-71 - Reference:

Simon RM, Subramanian J, Li MC, Menezes S. Using cross-validation to evaluate predictive accuracy of survival risk classifiers based on high-dimensional data. Brief Bioinform. 2011 May;12(3):203-14. doi: 10.1093/bib/bbr001. Epub 2011 Feb 15. PMID: 21324971; PMCID: PMC3105299.

Row 184 - Reference:

Rondeau, Virginie & Marzroui, Yassin & Gonzalez, Juan. (2011). frailtypack: An R Package for the Analysis of Correlated Survival Data with Frailty Models Using Penalized Likelihood Estimation or Parametrical Estimation. J. Stat. Softw.. 47. 10.18637/jss.v047.i04.

Moreover: 

 Rows 160-182 : A visual representation could help your (or your reader's) brain to quickly understand data and spot patterns, exceptions or outliers. For example a graphs also make it easier to illustrate relationships between data sets and provinces.

Row 306 - Reference of secondary data sourced from Statistics South Africa made readily available for researchers to use for their own research.

 Discussion: Try to explain "What do your results mean?", and  “how they relate to the literature”.

Conclusions: : Re-state the main points in a new concise way that you want your readers to remember.

way that you want your readers to remember.

Author Response

Dear Reviewer,

We sincerely appreciate your review of our manuscript. Your insightful suggestions have significantly improved the quality of our work, and we are truly grateful for your efforts.

Please see the attachment for our point-by-point responses to your feedback, along with the corresponding revised version of our paper, which is already uploaded in the MDPI system.

Thank you once again for your valuable input.

Reviewer 2 Report

The author has well-defined and comprehensive research problems. However, the authors’ current approach only focuses on large samples with heavy censoring. The reader is interested in seeing the performance of the author’s procedures on finite samples (especially small samples) under various censoring proportions. To improve the manuscript, the reader suggests that the author carry out simulations for various sample sizes and censoring proportions using distributions commonly used in lifetime data analysis. By looking at the simulation results (e.g., coverage probability and the length of the interval), the authors may be able to evaluate the performance of the procedures in practice.

In other words, the reader is asking the author to do more experiments to test the robustness of their procedures under different conditions. The reviewer also suggests that the author use commonly used distributions in lifetime data analysis, so that the results can be more generalizable.

Author Response

(The authors gave the same response as above.)

Author Response

(The authors gave the same response as above.)

Reviewer 4 Report

Please see the attached review report. 

Minor editing of English language required.

Author Response

(The authors gave the same response as above.)

Round 2

Reviewer 4 Report

This revision looks good to me. Well done. Here are two more minor issues need to be addressed.

(1) The quality of Figure 1 is poor. It is blurry and must be improved. 

(2) Row 134, "the covariates Xi = x_i1, x_i2, ..., x_ik and Xj = x_j1, x_j2, ..., x_jk", the covariates should be written as vectors, not like the current notation. 

Author Response

Dear Reviewer,

Once again, we would like to express our heartfelt appreciation for your critical review and insightful suggestions, which have greatly enhanced the quality of our paper. Please find the attached 2nd round response file.

Kind regards;

Yehenew Kifle
